# Evaluation of drug therapy problems, medication adherence and treatment satisfaction among heart failure patients on follow-up at a tertiary care hospital in Ethiopia

Elham Seid[1], Ephrem Engidawork[1], Minyahil Alebachew[1], Desalew Mekonnen[2], Alemseged Beyene Berha[1]*

1 Department of Pharmacology and Clinical Pharmacy, School of Pharmacy, College of Health Sciences, Addis Ababa University, Addis Ababa, Ethiopia, 2 Department of Internal Medicine, School of Medicine, College of Health Sciences, Addis Ababa University, Addis Ababa, Ethiopia

* alembeyen98@gmail.com

## Abstract

### Background

Drug therapy problems (DTPs) are major concerns of healthcare and have been identified to contribute to negative clinical outcomes. The occurrence of DTPs in heart failure patients is associated with worsening of outcomes. The aim of this study was to assess DTPs, associated factors and patient satisfaction among ambulatory heart failure patients at Tikur Anbessa Specialized Hospital (TASH).

### Methods

A hospital based prospective cross-sectional study was conducted on 423 heart failure patients on follow up at TASH. Data was collected through patient interview and chart review. Descriptive statistics, binary and multiple logistic regressions were used for analyses and P < 0.05 was used to declare association.

### Results

Majority of the patients were in NYHA class III (55.6%) and 66% of them had preserved systolic function. DTPs were identified in 291(68.8%) patients, with an average number of 2.51 ±1.07.per patient. The most common DTPs were drug interaction (27.3%) followed by non-compliance (26.2%), and ineffective drug use (13.7%). β blockers were the most frequent drug class involved in DTPs followed by angiotensin converting enzyme inhibitors. The global satisfaction was 78% and the overall mean score of treatment satisfaction was 60.5% (SD, 10.5).

### Conclusion

Prevalence of DTPs as well as non-adherence among heart failure patients on follow up is relatively high. Detection and prevention of DTPs along with identifying patients at risk can

**Data Availability Statement:** All relevant data are within the manuscript and its Supporting Information files.

**Funding:** The source of fund is medication therapy management thematic project of School of Pharmacy Addis Ababa University. The funders have role in the study design, data collection and analysis, decision to publish, or preparation of the manuscript.

**Competing interests:** The authors have declared that no competing interests exist.

**Abbreviations: ACEIS**, Angiotensin converting enzyme inhibitors; **ADR**, Adverse drug reaction; **AHA/ACC**, American heart association American college of cardiology; **AR**, Aortic Regurgitation; **ARBs**, Angiotensin receptor blockers; **AS**, Aortic Stenosis; **CMP**, Cardiomyopathy; **CRVHD**, Chronic rheumatic valvular heart disease; **CVD**, Cardiovascular disease; **DDI**, Drug-Drug interaction; **DI**, Drug interaction; **DM**, Diabetes mellitus; **DNOs**, Drug related negative outcomes; **DRPs**, Drug related problems; **DTPs**, Drug therapy problems; **HF**, Heart failure; **HFmrEF**, Heart failure with mid-range ejection fraction; **HFpEF**, Heart failure with preserved ejection fraction; **HFrEF**, Heart failure with reduced ejection fraction; **HHD**, Hypertensive heart disease; **HTN**, Hypertension; **LVEF**, Left ventricular ejection fraction; **MGL**, Morisky Green Levin Medication Adherence Scale; **MR**, Mitral Regurgitation; **MRA**, Mineralocorticoid receptor antagonist; **MS**, Mitral Stenosis; **NYHA**, New York heart association; **RCT**, Randomized control trials; **RHD**, Rheumatic heart disease; **SSA**, Sub-Sahran Africa; **TASH**, Tikur Anbessa specialized hospital; **TR**, Tricuspid regurgitation; **UGIB**, Upper gastro intestinal bleeding; **WHO**, World health organization.

save lives, help to adopt efficient strategies to closely monitor patients at risk, enhance patient's quality of life and optimize healthcare costs.

## Background

Heart disease remains the major public health concern and leading cause of death worldwide. Cardiovascular diseases (CVDs) accounted for nearly 836,546 deaths in the USA, out of which 9% was due to heart failure (HF) [1]. According to the American Heart Association (AHA) projection, future direct medical costs of HF would increase from $31 billion in 2012 to $70 billion in 2030 [2]. Death due to CVDs also takes place in low- and middle-income countries and sub-Saharan Africa (SSA), contributing to 5.5% of the global CVD deaths [3]. Data from different parts of Ethiopia also showed that CVD was the leading cause of death from non-communicable disease [4, 5].

HF is caused by any structural or functional cardiac abnormality, resulting in impairment of ventricular filling or ejection of blood [6]. HF is caused by various etiologies, each requiring unique management. The vast majority of HF in SSA are due to non-ischemic causes [7].

Neuro-hormonal antagonists and evidence-based beta-blockers have been shown to improve survival in patients with HF-with reduced ejection fraction (HFrEF) [8]. However, no treatment has yet been convincingly shown to reduce morbidity or mortality in HF with preserved ejection fraction (HFpEF) or HF with mid-range ejection fraction (HFmrEF) patients. In fact, the management of HFpEF is focused on managing congestion and comorbid conditions [9].

The care of HF patients is commonly complicated by the presence of comorbidity and poly-pharmacy, which in turn intensify risk of occurrence of drug therapy problems (DTPs) [10, 11]. A DTP is any undesirable event or circumstance experienced by a patient that involves, or is suspected to involve, drug therapy, and that interferes with achieving the desired goals of therapy. [12]. The occurrence of DTP may compromise treatment effectiveness and reduce quality of life [13]. Various studies showed that DTPs are the dominant reason for hospital admission and emergency department visits [14–16]. HF patients are at high risk of having DTPs [17] and frequencies as high as 78% have been reported [18]. The increased number of drugs prescribed has an important impact on HF patients, as it is associated with frequent hospitalization, waste of resources, adverse drug events, potential drug-drug interactions, and poor patient compliance [11, 19–21].

Patient satisfaction is an important measure of healthcare quality, as it offers information on the provider's success in meeting clients' expectations [22]. Low patient satisfaction may result in loss of trust and consequently in changing treating physicians or healthcare facilities or even discontinuing treatment [23]. Evaluating to what extent patients are satisfied with health services is clinically relevant, as satisfied patients are more likely to comply with treatment [24].

Optimization of drug therapy and prevention of DTPs are major factors for improving health care, reducing expenditure, and saving lives [25]. In practice, most HF patients are treated as outpatients, and their care in this setting is challenging, because less time is available for outpatient evaluation, and much more reliance is placed on appointment visits. Early identification of DTPs therefore helps to prevent and manage them through developing a better care plan. Studies regarding DTPs and patient satisfaction in ambulatory HF patients are

limited in Ethiopia. Thus, the present study aimed to assess DTPs and patient satisfaction among ambulatory HF patients in a tertiary care teaching hospital of Ethiopia.

## Methods

### Study setting

The study was conducted in the adult cardiac clinic of Tikur Anbessa specialized Hospital (TASH), the largest tertiary care teaching hospital of the country, with 700 bed capacity. Adult cardiac clinic is one of the many chronic follow-up clinics of the hospital giving service four days per week and manned by cardiologists, cardiology fellows, residents and nurses.

### Patients' involvement

Patients did not participate in conception and design of the study. However, they provided input during the pre-test (5% of the sample size), which helped us to make changes in the patient approach and timing for an interview during data collection. Patients played a central role in determining the level of medication adherence and treatment satisfaction.

### Study design and study period

A prospective cross-sectional study design was employed and carried out from 20 June to 20 August 2017 at the adult cardiac clinic of TASH

### Study population and inclusion criteria

Participants were all HF patients on follow up during the study period and who fulfilled the inclusion criteria. Patients aged ≥14 and diagnosed with HF, on active follow up and receiving treatment for at least 6 months, and had complete medical records including echocardiography findings were included. Patients with decompensated mitral stenosis (MS), missing (incomplete) data and not willing to participate were excluded.

### Sample size determination and sampling technique

Sample size (n = 423) was calculated using single population proportion formula. Since there was no previous study done on DTP among HF patients in Ethiopia, the proportion (P) of DTP in HF patients was taken as 50%. Systematic random sampling was used to select study participants. Following determination of sampling fraction, the first patient was selected randomly and every 3rd patient was included in the study. For those who had repeated clinic visits during the study period, data were collected only during their first visit.

### Data collection management

Structured questionnaire was used by two nurses for patient interview to obtain demographic data and clinical characteristics related to duration of treatment, frequency of hospital admission and current symptoms. Two clinical pharmacists reviewed patient charts for etiology of HF, echocardiographic parameters, NYHA functional class, comorbidities, history of allergies, vital signs, pertinent laboratory values, relevant past medical and medication history and current medications. Key informant interview was also administered for 6 physicians.

Appropriateness of medical treatment was evaluated using most updated guidelines of European Society of Cardiology and American Heart Association. The last three records of NYHA class, blood glucose and blood pressure were considered for identification of DTPs. Drug-drug interactions (DDIs) were evaluated using Micromedex® (Micromedex 2.0, Truven

Health Analytics Inc.) and only absolute contraindications and major drug interaction were considered. The identified DTPs were classified using Cipolle *et al*. (2012) DTP registration format.

Level of adherence was separately assessed using the Morisky Green Levin Medication Adherence Scale (MGL). Thus, patients were considered adherent if the score was ≤2 and non-adherent if the score was >2. Patient satisfaction was assessed using Treatment Satisfaction with Medicines Questionnaire (SATMED-Q) composed of 17 items with 6 (domains) dimensions. Satisfaction with each of the specific domains was scored using a five-point Likert scale and a score of >50 was considered as satisfied.

Pretest on 5% of the sample size was performed to check for uniformity and understandability of the data collection instrument and the tool was modified accordingly. One day training was given to data collectors and daily supervision was carried out to ensure accuracy and consistency of collected data.

### Data analysis

The data was entered into EPI-info V.7.2.1 and analyzed using Statistical package for social sciences (SPSS) version 21. Descriptive statistics was used to summarize patient characteristics. Categorical variables were described by frequencies and percentages and continuous variables by mean and standard deviations. A bivariate analysis was performed to identify determinants and variables with a p value<0.25 were included in the multivariate analysis to control confounding factors. Odds ratio (OR) with 95% confidence interval was computed for each variable and a p value <0.05 was considered as significant. Drug risk ratio (frequency of involvement in DTP divided by frequency of prescription) was used to identify drugs with high risk for creating DTPs.

### Ethical considerations

Ethical clearance was obtained from the Ethical Review Committee of School of Pharmacy as well as the Research and Ethics Committee of Department of Internal Medicine, School of Medicine, College of Health Sciences, Addis Ababa University. Participants were informed about the objective, benefit and risk of the study. This was followed by obtaining a written consent (for ≥18 years old) and assent as well as consent from the parent/guardian for those <18 years of age. Confidentiality and privacy were maintained through anonymity and restricting data access. The proposal including Amharic written verbal consent, which was attached as an annex was submitted to the Institutional Review Board (IRB) for approval. The study was conducted after obtaining a letter of approval from the School of Medicine (AAUMF 03–008) as well as the School of Pharmacy (002/17/SPharma) Boards.

## Results

### Demographic characteristics of patients

Mean age of the patients was 46.52±17, with a range of 14 and 90 years. The female gender was relatively high (53%) than the male gender. About two-third (64%) of them were married and about one-third (31.4%) had no formal education. Majority of the patients (90.5%) didn't use any of the social drugs and more than half (54%) of them paid out of pocket for their medications (Table 1).

**Table 1. Socio-demographic characteristics of heart failure patients attending at the cardiac clinic of Tikur Anbessa Specialized Hospital, Addis Ababa, Ethiopia, June—August, 2017.**

| Socio demographic | Category | Number (%) | Mean ± SD | Range |
|---|---|---|---|---|
| **Gender** | Male | 200 (47.3) | | |
| | Female | 223 (52.7) | | |
| **Age** | 14–35 | 111 (26.2) | | |
| | 36–60 | 2 14(50.6) | 46.52±17 | 14–90 |
| | >60 | 98 (23.2) | | |
| **Marital status** | Single | 105 (24.8) | | |
| | Married | 270 (63.8) | | |
| | Divorced | 17 (4) | | |
| | Widowed | 31 (7.3) | | |
| **Educational status** | No formal education | 133(31.4) | | |
| | Primary | 126(29.8) | | |
| | Secondary | 120 (28.4) | | |
| | Diploma and above | 44 (10.4) | | |
| **Social drug use** | Cigarette Smoking | 3 (0.7) | | |
| | Alcohol use | 21(5) | | |
| | Khat chewing | 14 (3.3) | | |
| **Hospitalization in the past one year** | | 156(36.9%) | | |
| **Source of medication** | Free | 196 (46.3) | | |
| | Paid | 227 (53.7) | | |

## Clinical characteristics of patients

The commonest cause of HF was chronic rheumatic valvular heart disease (CRVHD) (50.8%). Isolated valves were involved in 108 (50.2%) and multiple valves in 107 (49.8%) of the cases. A combination of mitral stenosis (MS) and mitral regurgitation (MR) was observed in 48 patients, while 40 patients had an isolated MR (Fig 1). Mean duration of treatment for HF was 6 ±4.6 years. Over half (55.6%) of the patients were in NYHA Class III and two-third (66.2%) of them had a preserved systolic function. Almost all (98.8%) patients reported no experience

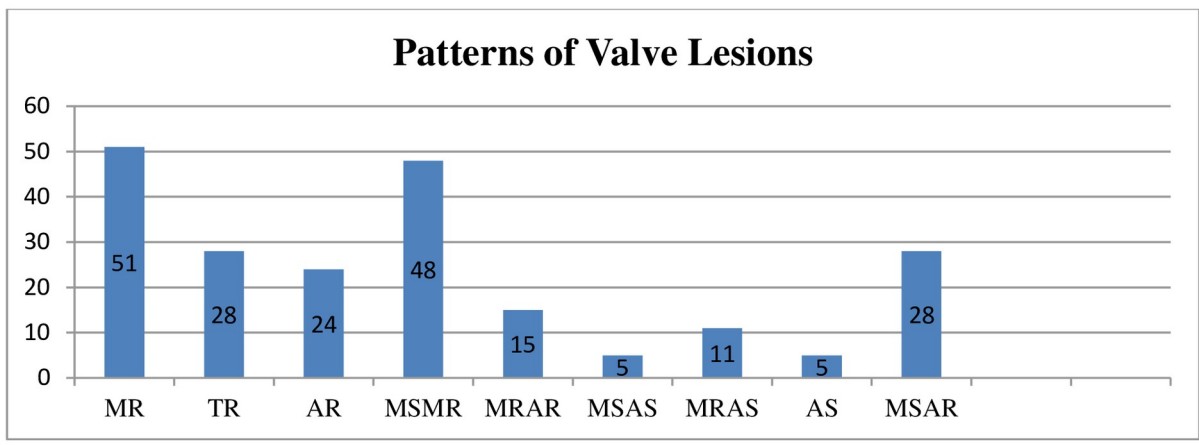

**Fig 1. Common valve lesions observed among heart failure patients attending at the cardiac clinic of Tikur Anbessa Specialized Hospital, Addis Ababa, Ethiopia June—August, 2017.** AR: Aortic Regurgitation, AS: Aortic Stenosis, MR: Mitral Regurgitation, MS: Mitral Stenosis, TR: Tricuspid Regurgitation.

**Table 2. Clinical characteristics of heart failure patient attending at the cardiac clinic of Tikur Anbessa Specialized Hospital, Addis Ababa, Ethiopia June—August, 2017.**

| Clinical CXS | Category | Number (%) | Mean ± SD | Range |
|---|---|---|---|---|
| Cause of heart failure | CRVHD | 215(50.8) | | |
| | IHD | (91)21.5 | | |
| | HHD | 62(14.7) | | |
| | CMP | (45)10.6 | | |
| | Others | (10)2.4 | | |
| Duration of heart failure treatment | ≤5years | 235(55.5) | | |
| | 6-10yrs | 135(32) | 6 ±4.6 | 6month-35yrs |
| | >10yrs | 53(12.4) | | |
| Frequency of follow up | ≤3 month | 316(74.7) | | |
| | ≥4 month | 107 (25.3) | | |
| NYHA class | Class I | 63 (14.9) | | |
| | Class II | 125 (29.6) | | |
| | Class III | 235 (55.6) | | |
| LVEF | HFrEF (<40%) | 104 (24.9%) | 55.43±14.4 | 15–81 |
| | HFmrEF (40–49%) | 39 (9.2%) | | |
| | HFpEF (≥50%) | 280 (66.2%) | | |
| Known drug allergy | No | 418 (98.8) | | |
| | Yes | 5 (1.2) | | |
| Comorbidity | Hypertension | (173)40.9 | | |
| | Atrial fibrillation | (158)37.4 | | |
| | Diabetes mellitus | (67)15.8 | | |
| | Dyslipidemia | (26)6.1 | | |
| | Asthma | (21)5 | | |
| | Stroke | (18)4.3 | | |
| | Peripheral neuropathy | (11)2.6 | | |
| | Others | (38)9 | | |

*Other: Degenerative valvular heart disease, corpulomonale, congenital heart disease; CMP: cardiomyopathy, CRVHD: chronic rheumatic valvular heart disease, IHD: ischemic heart disease, HHD: hypertensive heart disease, HFrEF: heart failure with reduced ejection fraction, HFmrEF: heart failure with mid-range ejection fraction, HFpEF: heart failure with preserved ejection fraction, LVEF: left ventricular ejection fraction.

of drug allergy. Comorbidity was found in 318 (75.2%) of the patients, with hypertension and atrial fibrillation accounting for 40.9% and 37.4%, respectively (Table 2).

Patient characteristic was related to causes of HF and disaggregation revealed that patients with CRVHD (33.7%) classified mainly under the ages of 36–60 years and IHD (12.0%) for > 60 years (Table 3).

In an attempt to look at the contribution of the different etiologies to HF with time, the data was disaggregated. The analysis showed that 235 (55.5%) patients had had HF for ≤ 5 years, out of which 48.5% had CRVHD, 22.9% ischemic heart disease (IHD), and 13.6% hypertensive heart disease (HHD). The number of patients having the three etiologies decreased with duration HF (Table 4).

A large proportion (63.4%) of patients received 5–10 drugs per day and mean number of drugs per day was 4.96±1.62 per patient. A total of 2097 medications were prescribed and diuretics (27.6%) and beta blockers (14.5%) were the commonly prescribed drug classes. The most frequently prescribed specific drugs were furosemide (294), spironolactone (236), enalapril (188), benzathine penicillin (181), aspirin (171) and atenolol (170) (Fig 2).

**Table 3. Disaggregation of type of patient characteristics with causes of heart failure in patients attending at the cardiac clinic of Tikur Anbessa Specialized Hospital, Addis Ababa, Ethiopia June—August, 2017.**

| Variables | | CRVHD | IHD | HHD | CMP | Others |
|---|---|---|---|---|---|---|
| Age | 14–35 | 16 | 1 | - | 10 | 1 |
| | 36–60 | 98 | 42 | 15 | 20 | 2 |
| | >60 | 12 | 35 | 22 | 13 | 4 |
| Gender | Male | 31 | 48 | 15 | 21 | 1 |
| | Female | 95 | 30 | 22 | 22 | 6 |
| Systolic BP | <120 | 119 | - | - | 25 | 7 |
| | 120–129 | 7 | 72 | 16 | 12 | - |
| | 130–139 | - | 5 | 14 | 6 | - |
| | ≥140 | - | 1 | 7 | - | - |
| Comorbidity | Yes | 117 | 76 | 37 | 37 | 6 |
| No of medication | <5 | 20 | 10 | 7 | 6 | 4 |
| | ≥5 | 106 | 68 | 30 | 37 | 3 |

CMP: cardiomyopathy, CRVHD: chronic rheumatic valvular heart disease, IHD: ischemic heart disease, HHD: hypertensive heart disease.

## Prevalence and type of DTPs

A total of 730 DTPs were identified in 291(68.8%) patients. One DTP was identified in 46 (15%), 2 in 85 (29.2%), three in 93 (31.9%) and ≥ 4 in 67(23%) patients. The average number of DTP per patient was 2.51±1.068.

The most common DTPs identified were drug-drug interactions (DDI) (27.3%) followed by noncompliance (26.2%), ineffective drug (13.7%), adverse drug reaction (ADR) (11.5%) and need additional drug therapy (9.4%) (Table 5). The most frequent drug classes involved in DTPs were beta blockers followed by angiotensin converting enzyme inhibitors (ACEIs), and mineralocorticoid receptor antagonist (MRA) (Fig 3)

Prevalence of DTP was related to causes of HF and disaggregation revealed that patients with CRVHD (38.6.%) had the highest DTP followed by IHD (32.8%) (Table 6).

A total of 199 significant DDIs were identified. At least one drug interaction (DI) was found in 141, 2 in 42, and more than two in 16 patients. The commonest DDI found was the use of spironolactone and digoxin with arbitrary monitoring of potassium level. Such use might have resulted in increased digoxin toxicity.

**Table 4. Disaggregation of duration of heart failure with etiology among patients attending at the cardiac clinic of Tikur Anbessa Specialized Hospital, Addis Ababa, Ethiopia June—August, 2017.**

| Etiology of HF | Duration of heart failure | | | |
|---|---|---|---|---|
| | ≤ 5years | 6–10 years | >10 years | Total |
| CRVHD | 114 (53%) | 74(34.4%) | 27(12.6%) | 215 |
| IHD | 54 (59.3%) | 30 (32.9%/) | 7 (7.7%) | 91 |
| HHD | 32 (51.6%) | 15 | 15 | 62 |
| CMP | 31 (68.9%) | 11 (24.4%) | 3 (6.7%) | 45 |
| Others | 4 (40%) | 5 (50%) | 1 (10%) | 10 |
| Total | 235 | 135 | 53 | 423 |

CMP: cardiomyopathy, CRVHD: chronic rheumatic valvular heart disease, IHD: ischemic heart disease, HHD: hypertensive heart disease.

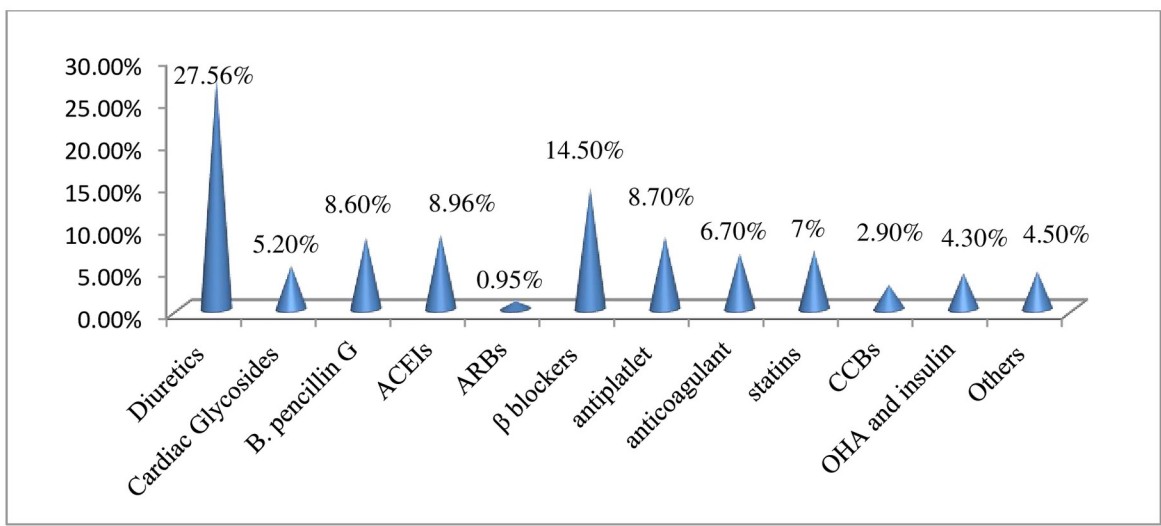

**Fig 2. Frequently prescribed drug class in heart failure patients attending at the cardiac clinic of Tikur Anbessa Specialized Hospital, Addis Ababa, Ethiopia June-August, 2017.** Anti arrhythmic, antianginal agent, antithyroid, antiTB, HAART, Iron salt, NSAIDs, xanthine oxidase inhibitor, tricyclic antidepressant; ACIEs: angiotensin converting enzyme inhibitors, ARBs: angiotensin receptor blockers, CCBs: calcium channel blockers, OHA: oral hypoglycemic agents.

About 20% (n = 84) of patients experienced medication adverse effects. These included gum bleeding due to warfarin (22); dry cough (20) and angioedema (1) due to ACEIs; and peripheral edema due to nifedipine (11). Other ADRs reported were penicillin allergy, upper gastrointestinal bleeding (UGIB) due to dual antiplatelet therapy, atenolol induced bradycardia and bronchospasm & hypotension due to high dose of furosemide, myopathy related to atorvastatin and spironolactone induced gynecomastia and hyperkalemia.

Drug risk ratio was calculated for each drug. A higher drug risk ratio was found for Nifedipine (0.7) followed by Metoprolol tartrate (0.6), Atenolol (0.33) and Enalapril (0.25). Although furosemide was the most frequently used, it was found to have a lowest drug risk ratio (0.07).

According to the MGL medication adherence scale, 191 (26.2%) of patients were non-adherent to their medication. Reason for non-adherence included forgetfulness (43.5%),

**Table 5. Type of DTP identified from heart failure patients attending at the cardiac clinic of Tikur Anbessa Specialized Hospital, Addis Ababa, Ethiopia June—August, 2017.**

| Type of DRP | Specific DTP | No | Percent |
|---|---|---|---|
| Drug interaction | | 199 | 27.3 |
| Adverse drug reaction | Undesired effect | 74 | 11.5 |
| | Unsafe drug for the patient | 5 | |
| | Allergic reaction | 5 | |
| Effectiveness | Ineffective drug | 100 | 13.7 |
| Inappropriate dose | Suboptimal dose | 52 | 7.1 |
| | Dose high | 4 | 0.55 |
| Need additional drug therapy | Prophylactic | 55 | 7.5 |
| | Synergy | 14 | 1.92 |
| Unnecessary drug therapy | No medical indication | 17 | 2.3 |
| Compliance | Non drug therapy | 14 | 1.92 |
| | Non compliance | 191 | 26.2 |

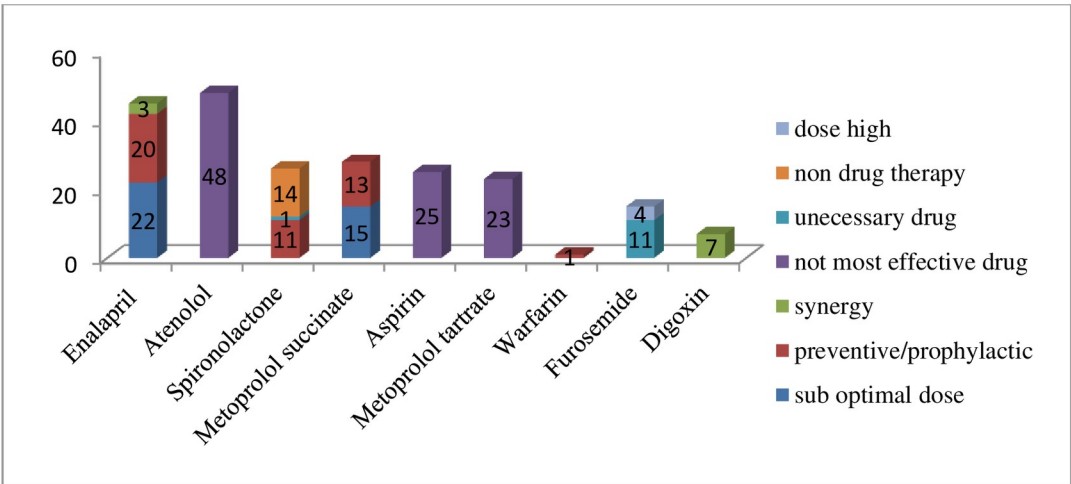

**Fig 3. Drug classes involved in specific type of drug therapy problems among heart failure patients attending at the cardiac clinic of Tikur Anbessa Specialized Hospital, Addis Ababa, Ethiopia June—August, 2017.**

regimen complexity (40%), unavailability of medication (22.3%), worsening of symptoms (19.6%), drug side effect (19.3%) and cost of medication (14.9%).

## Predictors of DTP

Variables with p value < 0.25 in the bivariate analysis and included in the multivariate logistic regression were age, gender, marital status, educational status, NYHA class, left ventricular ejection fraction (LVEF), presence of comorbidity, duration of HF treatment, source of medication and number of medications. Subsequent analysis with multivariable logistic regression revealed that age, NYHA class, LVEF, presence of comorbidity, duration of HF treatment, source of medication and average number of drugs/day were significantly associated with DTP (Table 7). Patients aged between 36–60 years were 4.8 times more likely (AOR = 4.8, 95% CI 1.804–12.806) to develop DTPs compared to patients with <35 years of age. NYHA class III patients had 5.6 times more chance (AOR = 5.65, 95% CI 2.714–11.768) to develop DTPs

**Table 6. Disaggregation of type of drug therapy problems with causes of heart failure in patients attending at the cardiac clinic of Tikur Anbessa Specialized Hospital, Addis Ababa, Ethiopia June—August, 2017.**

| Variables | CRVHD | IHD | HHD | CMP | Others |
|---|---|---|---|---|---|
| **Drug therapy problems** | | | | | |
| No medical indication | - | 14 | 2 | 1 | - |
| Non drug therapy more appropriate | 13 | - | - | - | 1 |
| Prophylactic/preventive drug therapy | 15 | 25 | 3 | 12 | - |
| Synergistic | 7 | - | 7 | - | - |
| More effective drug available | 23 | 50 | 5 | 18 | 4 |
| Dose low | 3 | 29 | - | 20 | - |
| Dose high | 4 | - | - | - | - |
| Adverse drug reactions | 36 | 22 | 21 | 4 | 1 |
| Drug interaction | 95 | 51 | 19 | 31 | 3 |
| Non compliance | 85 | 49 | 28 | 25 | 4 |

CMP: cardiomyopathy, CRVHD: chronic rheumatic valvular heart disease, IHD: ischemic heart disease, HHD: hypertensive heart disease.

**Table 7. Predictors of occurrence of drug therapy problems in heart failure patients attending the cardiac clinic of Tikur Anbessa Specialized Hospital, Addis Ababa, Ethiopia June—August, 2017.**

| Variables | Category | DTPs (%) | | COR (95% CI) | AOR (95% CI) |
|---|---|---|---|---|---|
| | | Yes | No | | |
| Gender | Male | 115(39.5) | 85(64.4) | 1.00 | 1.00 |
| | Female | 176(60.5) | 47(35.6) | 2.76(1.806–4.241) | 1.54(0.737–3.216) |
| Age | 14–35 | 28(9.6) | 83(62.9) | 1.00 | 1.00 |
| | 36–60 | 178(61.2) | 36(27.3) | 14.65(8.386–25.617) | 4.80(1.804–12.806) * |
| | >60 | 85(29.2) | 13(9.8) | 19.38(9.397–39.978) | 3.45 (1.017–11.71) * |
| Marital status | Ever married | 232(79.7) | 86(65.2) | 1.00 | 1.00 |
| | Single | 59(20.3) | 46(34.8) | 0.47(0.301–0.752) | 2.015(0.793–5.120) |
| Educational | Primary or less | 185(63.6) | 74(56.1) | 1.368(0.900–2.079) | 2.138(0.949–4.817) |
| | Secondary and above | 106(36.4) | 58(43.9) | 1.00 | 1.00 |
| NYHA class | Class I-II | 76(26.1) | 112(84.8) | 1.00 | 1.00 |
| | Class III | 215(73.9) | 20(15.2) | 15.84(9.205–27.265) | 5.65(2.714–11.768) * |
| Left ventricular ejection fraction | HFpEF | 195(67) | 124(93.9) | 1.00 | 1.00 |
| | HfrEF | 96(33) | 8(6.1) | 7.63(3.585–16.244) | 6.23(2.390–16.245) * |
| Comorbidity | No | 18(6.2) | 87(65.9) | 1.00 | 1.00 |
| | Yes | 273(93.8) | 45(34.1) | 2.93(2.864–11.386) | 5.23(1.032–7.196)* |
| Source of med | Paid | 147(50.5) | 80(60.6) | 1.00 | 1.00 |
| | Free | 144(49.5) | 52(39.4) | 1.507(0.992–2.289) | 2.32 (1.105–4.860)* |
| Duration of HF | ≤5yrs | 128(44) | 107(81nm.1) | 1.00 | 1.00 |
| | 6–10 | 114(39.2) | 21(15.9) | 4.538(2.667–7.722) | 1.097(1.009–1.193) * |
| | >10 | 49(16.8) | 4(3) | 10.24(3.580–29.29) | 4.415(1.190–16.375) * |
| Number of medications | <5 | 47(16.2) | 108(81.8) | 1.00 | 1.00 |
| | ≥5 | 244(83.8) | 24(18.2) | 23.36(13.595–40.145) | 7(3.263–15.280)* |

* Statistically significant at P-value < 0.05; AOR = Adjusted odds ratio; COR = Crude odds ratio, HFpEF: heart failure with preserved ejection fraction, HFrEF: heart failure with reduced ejection fraction.

compared to NYHA class I or class II patients. Patients with reduced ejection fraction were 6.2 times more likely (AOR = 6.23, 95% CI 23.90–16.245) to develop DTPs than patients with preserved ejection fraction. The odds of DTP among patients with one or more co-morbidities was 5.2 times higher (AOR = 5.23, 95% CI 1.032–7.196) than those with no co-morbidities. The odds of DTPs were 7 times higher (AOR = 7, 95% CI 3.263–15.280) among patients taking an average of 5 or more drugs per day as compared to those less than five drugs per day.

## Patient satisfaction

The satisfaction score for undesirable side effect and treatment effectiveness was 79% and 70%, respectively. However, the score for impact on daily activity was relatively lower, standing at 62%. The global satisfaction score was 78% and the overall mean score of treatment satisfaction was 60.5(SD, 10.5) (Table 8).

## Discussion

HF patients are at high risk of having DTPs and adherence issues due to comorbidity, polypharmacy and complexity of drug regimens. The presence of DTPs in patients with HF is associated with detrimental health outcomes. Identification of type of DTP and factors associated with them is therefore critical for prevention of DTPs and improving health outcomes.

**Table 8. The treatment satisfaction of heart failure patients attending cardiac clinic of Tikur Anbessa Specialized Hospital, Addis Ababa, Ethiopia June—August, 2017.**

| SATMED-Q dimension | Satisfaction score (Mean ± SD) |
|---|---|
| Undesirable side effects (0–100) | 79.3± 27.2 |
| Treatment effectiveness (0–100) | 70.5 ± 14.0 |
| Convenience of use (0–100) | 59.9 ± 20.6 |
| Impact on daily activities (0–100) | 70.6 ± 16.2 |
| Medical care (0–100) | 63.5 ± 19.2 |
| Global satisfaction (0–100) | 78.7 ± 20.1 |
| Total composite score (0–100) | 60.5 ± 10.5 |

There was a female preponderance in HF in the present study, which is in agreement with studies performed elsewhere [13, 19, 25]. HF appears to be more common in the adult population, as the mean age was 46.5 (SD 17) and 50.6% of them were in the age group of 36–60 years. This is in line with other local studies [26, 27]. As age increases, the number of medications and comorbidities were increased and also limit patients' physical activity. The number of comorbidities (75%) was found to be lower than other studies [28, 29]. The possible reason for this difference could be related to age difference of patients involved in the studies.

A total of 2097 drugs were used with a mean of 4.9 drugs per patient. This is comparable with some studies [28, 29] but different from other studies [17, 18], where a mean of >10 drugs/patient were prescribed. In the latter studies, more than 80% of the patients had comorbidities like hypertension and diabetes mellitus that required multiple therapies.

The prevalence of DTP obtained in the present study (69%) is in line with those reported in cardiovascular patients of India (66.3%) [30], Switzerland (69%) [31] and Gondar, Ethiopia (63.4%) [19]. However, it was lower than other rates reported from Barcelona (78%) [18] and Bonga, Ethiopia (72%) [32]. Difference in the study design (cohort vs cross sectional study), classification and operationalization of DTP, and nature of patients included in the study could explain the discordance.

Consistent to other studies [13, 18, 30], DDI was the most commonly encountered DTP. DDI is known to be a major factor affecting patient's clinical outcome by contributing to increased risk of adverse drug events related to hospitalization and a higher health care cost. Adherence related problem was the second most common DTP (26.2%) and found to be concordant with other rates reported India (28.9%) [33] and the Netherlands (28%) [34]. Higher rates than the current study had been reported from Cuba (36.5%) [35], Adama (Ethiopia) (44.8%) [13], Jimma (Ethiopia) (46.4%) [29] and Brazil (63.5%) [36]. Variation in the setting (outpatient vs. inpatient), methods used to measure adherence, or severity of the disease might account for the observed difference.

Problems related to drug effectiveness (13.7%) was the third most common DTP identified. Use of Aspirin instead of warfarin for prevention of stroke in patients with atrial fibrillation was one of the problems detected. However, guidelines state that aspirin is less effective in preventing stroke than warfarin [6]. Physicians cited different reasons for use of aspirin, including unavailability of warfarin, unaffordability and inaccessibility of INR monitoring. The other effectiveness problem was the use of Atenolol and Metoprolol tartrate in patients with HFrEF. The efficacy of Metoprolol tartrate in reducing mortality in HF has not been proven [2, 8]. Moreover, Metoprolol succinate provides more consistent plasma concentrations over a 24 h period compared to the immediate-release Metoprolol [37]. The reason for not using Metoprolol succinate was related to availability and cost.

ADRs were identified in 11.5% of the patients. A similar rate was reported from a Taiwan study (13.5%) [17], although higher rates were also reported from India (19%) [38] and Ethiopia (26%) [29]. Inappropriate dosing (suboptimal dose and dose high) accounted for 7.65% of all DTPs, where suboptimal dosing was observed in 7.1% of patients. This is concordant with studies done in Spain (6.7%) [28], India (10.4%) [30], and Ethiopia (7.5%) [29]. ACEIs and evidence based β blockers were found to be not titrated towards established guideline recommended target doses. Presence of other comorbidities, fear of adverse effects, reluctance to change the treatment when patients are stable, burden of monitoring, and lack of clinical experience were cited as reasons for failure to up-titrate the dose. On the other hand, dose high was observed only in four patients, unlike other studies [17, 29, 39], where higher rates were reported. The most common drug associated with high dose was furosemide, which led patients to experience hypotension.

Among indication related problems, the need for prophylactic/preventive drug therapy accounted for 7.5%, which was in line with studies conducted in Taiwan (7%) [17], Jordan (7.6%) [25] and India (9.4%) [38], those higher rates were reported from Barcelona (28.6%) [18] and Boston 43% [39]. Participants' level of education and awareness, the health care system and resources allocated, patient information regarding active disease, laboratory values and medications used would likely contribute the difference. Indeed, such factors could create opportunities for easy identification of DTPs and co-morbidities in HF patients. ACEIs and β blockers were among the most common drug categories needed for the prevention of cardiac remodeling and disease progression and this is in line with other local studies [29]. Unnecessary drug therapy was found in 4.2% of patients. Lower [25] and similar [18] rates to the current study have been reported in the literature.

Beta blockers, ACEIs, MRA and antiplatelet were among the common drug classes involved in DTP as reported elsewhere [17, 18, 30, 32]. It is essential to be aware of and give due attention to those drugs with the highest drug risk ratio, as they could expose patients more often to DTPs.

Identification of risk factors for DTPs is important to identify the most susceptible patients requiring close monitoring of drug therapy. Age, NYHA class, LVEF, presence of comorbidity, duration of HF treatment, source of medication and poly-pharmacy were found to be independent predictors DTPs. These findings are supported by a number of other studies [18, 19, 28, 38]. As age increases, comorbidity and number of medication increase, exposing patients to an increased risk for developing DTPs. Patients with reduced ejection fraction are at higher risk for developing DTPs. This could be because patients with reduced ejection fraction might not be on cardio protective medication (untreated medical condition) or may be on suboptimal dose.

Satisfaction with medication and medical treatment appears to be related to patient medication adherence, and constitutes a quality indicator that can be used for improving health care. Patient satisfaction influences the health behavior and would be crucial in patients with chronic diseases [40]. Global satisfaction of study participant was 78% and this congruent with scores reported from Greece (80%) [41] and Nigeria (78.6%) [42]. The overall mean score of treatment satisfaction (60.5%) is also a similar study [43] carried out in HF patients (over 60%). A lower score was found in the undesirable side effect domain (21%) and the score for satisfaction on convenience of use was also relatively lower (59%) compared to other dimensions. Lower score in the convenience domain compared to other dimensions is also reported [40] and might be attributed to pill burden that poses difficulty to adhere to dose regimens.

This study has some limitations. The cross-sectional nature of the study didn't allow follow up of patients and it was also a non-interventional study. No causal relationship had been established when ADRs were identified. Self- reporting was used to measure adherence, which

may suffer from recall bias, and hence could overestimate the level of adherence. In addition, the extent of generalisablity may be limited, since it was a single centered study conducted in a tertiary care hospital, where complicated and severe cases that might need combination therapy are referred. Nevertheless, the higher sample size, randomized sampling techniques, use of both patient interview and medical chart review could offset these limitations.

## Conclusions

Medication therapy plays a significant role in treating HF. However, optimal medication in HF outpatients remains to be a major challenge in clinical practice. This study provided evidence that several types of DTPs, including non-adherence are apparent among ambulatory HF patients and CRVHD appears to be the most common cause of the disease. Old Age, NYHA class, LVEF, comorbidity, duration of treatment, source of medication, and poly-pharmacy have been identified as predictors of DTPs. A relatively good global satisfaction of study participants was obtained. These issues should be appropriately addressed in order to improve the overall outcome of patients.

## Supporting information

**S1 Questionnaire.**
(DOCX)

**S1 Rawdata.**
(DOCX)

**S1 File.**
(DOCX)

## Acknowledgments

We are very thankful to the study participants for their willingness to participate and nursing staff of cardiac clinic for their cooperation in the data collection process.

## Author Contributions

**Conceptualization:** Elham Seid, Ephrem Engidawork, Minyahil Alebachew, Desalew Mekonnen, Alemseged Beyene Berha.

**Data curation:** Elham Seid, Ephrem Engidawork, Alemseged Beyene Berha.

**Formal analysis:** Elham Seid, Ephrem Engidawork, Alemseged Beyene Berha.

**Funding acquisition:** Ephrem Engidawork, Minyahil Alebachew, Alemseged Beyene Berha.

**Investigation:** Elham Seid, Ephrem Engidawork, Desalew Mekonnen, Alemseged Beyene Berha.

**Methodology:** Elham Seid, Ephrem Engidawork, Minyahil Alebachew, Desalew Mekonnen, Alemseged Beyene Berha.

**Project administration:** Ephrem Engidawork, Minyahil Alebachew, Alemseged Beyene Berha.

**Software:** Elham Seid, Ephrem Engidawork, Minyahil Alebachew, Alemseged Beyene Berha.

**Supervision:** Ephrem Engidawork, Desalew Mekonnen, Alemseged Beyene Berha.

**Writing – original draft:** Elham Seid, Ephrem Engidawork, Alemseged Beyene Berha.

**Writing – review & editing:** Elham Seid, Ephrem Engidawork, Minyahil Alebachew, Desalew Mekonnen, Alemseged Beyene Berha.

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
