## [Decision Letter · Decision Letter 0]

25 Feb 2020

PONE-D-19-34755

Evaluation of Drug Therapy Problems, Medication Adherence and Treatment Satisfaction among Heart Failure Patients on Follow-up at a Tertiary Care Hospital in Ethiopia

PLOS ONE

Dear Mr. Berha,

Thank you for submitting your manuscript to PLOS ONE. After careful consideration, we feel that it might have merit but does not meet PLOS ONE’s publication criteria as it currently stands. Therefore, we invite you to submit a revised version of the manuscript that addresses the points raised during the review process.

Please note that there are substantial changes required and your manuscript will only be considered if you can address them. In addition to the reviewer's comments, I would like you to particularly consider the following points:

- The definition of DTP must be clear and some need to be reconsidered (see also the reviewer's comment). Thus, a list with all considered DTP should be given (as supplementary table for instance). In addition, you need to carefully reconsider what a DTP is. E.g. interactions between medication is used as DPT that are recommended by the guidelines (combination of ACE-inhibitor and spironolactone). In fact, all top 5 DDIs are often used in clinical practice and not considered as clinically relevant. This can hardly be considered as DTP.

- I would suggest to divide clearly between HFrEF (you may include HFmrEF if you want) and the rest because the recommendation are different. 

- The discussion should be shortened and focused significantly.

- You need to mention the statistical analyses used.

We would appreciate receiving your revised manuscript by Apr 10 2020 11:59PM. To enhance the reproducibility of your results, we recommend that if applicable you deposit your laboratory protocols in protocols.io, where a protocol can be assigned its own identifier (DOI) such that it can be cited independently in the future. For instructions see: http://journals.plos.org/plosone/s/submission-guidelines#loc-laboratory-protocols

We look forward to receiving your revised manuscript.

Kind regards,

Hans-Peter Brunner-La Rocca, M.D.

Academic Editor

PLOS ONE

Journal Requirements:

2. Please state in your manuscript whether you had permission to use the Morisky Medication Adherence Scale-4.

4. Ethics Statements should appear in the Methods section of the manuscript. In particular, we want to ensure that ethics statements are not in a separate section at the end of the manuscript.

In the manuscript file, scroll down to the references and check if there are any ethic statements in the Acknowledgements or in a separate section at the end of the manuscript (after Discussion and/or Conclusion sections).

Reviewers' comments:

Reviewer's Responses to Questions

**Comments to the Author**

1. Is the manuscript technically sound, and do the data support the conclusions?

Reviewer #1: Partly

2. Has the statistical analysis been performed appropriately and rigorously? 

Reviewer #1: I Don't Know

3. Have the authors made all data underlying the findings in their manuscript fully available?

Reviewer #1: No

4. Is the manuscript presented in an intelligible fashion and written in standard English?

Reviewer #1: No

5. Review Comments to the Author

Reviewer #1: The authors provide a cross-sectional analysis of DTPs in a subset of patients with presumed heart failure and conclude that the prevalence of DTPs and non-adherence among HF patients is high.

I have a number of comments:

1) The authors state that the commonest cause of HF was chronic rheumatic valvular heart disease (CRVHD); namely in 50.8%. Please specify further: how many were mitral valve stenosis? how many were mitral valve regurgitation? or both? how many were aortic valve stenosis? etc.

This is of crucial importance because distinct myocardial structural and/or functional abnormalities will develop according to which type of valve is affected. For instance, in the setting of mitral valve stenosis, the LV is protected from pressure or volume loading and signs and symptoms of HF result from high pulmonary pressures without elevated ventricular pressures. Furthermore, patients with decompensated mitral valve stenosis do not fit in the chronic HF guideline drug therapy recommendations and therefore should not be included in this study. Indeed, these patients should receive diuretics, possibly rate control for atrial fibrillation, but above all mitral valve intervention.

Therefore, to obtain a more relevant (chronic HF) patient population, analyses should be more specific according to type of CRVHD.

2)If HF results from VHD, then a class I indication exists for valvular intervention. Please specify the reasons why this has not been performed.

3)Please specify the length of HF duration according to etiology of HF.

4)Generalizability is limited because of the relatively young age and high prevalence of CRVHD.

5)The authors state a prevalence of HFpEF of 66.2% in their group. However, this will be enriched with CRVHD patients (probably MV stenosis). This cannot be classified as HFpEF and should be changed.

6)Specify the reason for inappropriate dose (Table 3.): this could be because of uptitration phase, hypotension, renal insufficiency, bradycardia etc. Hence, this should not be recorded as a DTP perse.

7)Anecdotic interactions are scored as DTPs (such as diuretic with aspirin; or spironolactone with aspirin etc). In the setting of HF due to ischemic heart disease, these combinations are class I recommended. This should not be regarded as a DTP or DDI. Please adjust the data accordingly.

8)In the discussion the authors frequently refer to other papers. The HF populations described in these references however differ substantially in their characteristics with the currently studied patient population. Again, generalizability is limited, because of the very high prevalence of untreated CRVHD and young patient age.

9)The authors claim that the most frequent DDI was between aspirin and furosemide with increased risk of nephrotoxicity and reduced diuretic effectiveness. This again is anecdotic. Please provide data about the actual occurrence of aspirin/furosemide induced nephrotoxicity and reduced diuretic effectiveness.

10)Dual antiplatelet therapy is class I recommended in ACS patients. Therefore, this should not be regarded as a DDI. Please adjust. It is more relevant to depict the DTP's with actual relevant clinical impact.

11)Expand Table 4 with patient characteristics, such as age, comorbidities, gender, renal function (eGFR), blood pressure and medication

12)How many patients have been hospitalized for HF in the past?

13)Please have the manuscript reviewed by a native English speaker.

6. PLOS authors have the option to publish the peer review history of their article (what does this mean?). If published, this will include your full peer review and any attached files.

Reviewer #1: No

---

## [Author Response · Author response to Decision Letter 0]

10 Jul 2020

Reviewers’ Comments and Authors Response

Manuscript Number: PONE-D-19-34755

Manuscript Title: Evaluation of Drug Therapy Problems, Medication Adherence and Treatment Satisfaction among Heart Failure Patients on Follow-up at a Tertiary Care Hospital in Ethiopia

Authors: Elham Seid, Ephrem Engidawork, Desalew Mekonnen, Minyahil Alebachew, Alemseged Beyene

We would like to thank the Editor and reviewers for their critical comments and constructive suggestions, which contributed to the quality of the manuscript. We have revised the manuscript as advised and changes made are highlighted in yellow. Point-by-point response for the reviewers’ comment is given below. We hope the revision undertaken has improved the manuscript to a level of your satisfaction and we request your editorial hand. 

Response to Editorial Comments 

Comment 1: Definition of DTP with supplementary table 

Response 1: Accommodated. 

Comment 2: Drug interaction as DTP since all top 5 drug interaction are clinically not relevant (hardly considered as DTP )

Response 2: Removed all the others except those associated with adverse drug reactions. 

Comment 3: Clearly divide between HFrEF and HFmrEF

Response 3: Accommodated.

Comment 4: The discussion should be short and focused significantly 

Response 4: Shortened as much as possible 

Comment 5: Permission for MMA4

Response 5:- Since we did not hear from the License holder, we used the equivalent scale “the four –item Morisky Green Levin Medication Adherence Scale (MGL)” , which is a free online version. 

Comment 6: Mention statistical analysis used 

Response 6: We elaborated the statistical analysis used and also included in the abstract. 

Response to Reviewer' comments 

Comment 1: Please specify further the types of valve involved in CRVHD patients 

Response 1: The types of valve lesion involved are provided in a new Figure (Figure 1). Please take note that patients with decompensated MS were not included in the study. 

Comment 2: If HF results from VHD, then a class I indication exists for valvular intervention. Please specify the reasons why this has not been performed.

Response 2: Valvular surgery was not provided when the study was initiated. It used to done through mission based aid or sponsor based travel abroad. 

Comment 3: Please specify the length of HF duration according to etiology of HF.

Response 3: Accommodated (See Table 4).

Comment 4: Generalizability is limited because of the relatively young age and high prevalence of CRVHD

Response 4: Dear reviewer: I agree with your point, however since diagnosis in developing countries like Ethiopia is usually established at an advanced stage of the disease, more than half of the patients are above 35 years old. Moreover, almost half of the patients included in this study had heart failure caused by other than CRVRHD. 

Comment 5: A prevalence of HFpEF of 66.2% in their group. However, this will be enriched with CRVHD patients (probably MV stenosis).

Response 5: Dear reviewer: thank you for your valuable comment, but as it is indicated in specific valve involved; in this study patients with MS also have other valve lesions. Moreover, the classification of ejection fraction was taken from the patient’s medical chart. 

Comment 6: Specify the reason for inappropriate dose (suboptimal dose)

Response 6: The dose low is considered as DTP if patients with HFrEF and on ACEIs or beta blockers were not up titrated after carefully consideration of patient status and after excluding any comorbidity or contraindication and the patient at high risk of toxicity due to increased dose (patients who had an organ dysfunction). Unfortunately, as I have raised in the discussion part patients remain in the same dose for several years ( for example patients with IHD and EF of 34 might be on Metoprolol 12.5 mg for years)

Comment 7: Anecdotic interactions are scored as DTPs (such as diuretic with aspirin; or spironolactone with aspirin etc

Response 7: We have made a correction and removed such types of drug interactions

Comment 8: Generalisablity is limited, because of the very high prevalence of untreated CRVHD and young patient age.

Response 8: As per the suggestion; we have made a correction in the discussion part. The main problem is there were only a few studies available and even these studies were done outside Africa, where the cause is not RHD. However we have tried to use these studies only if the problem is almost identical. For example patient with reduced EF and not on cardio protective therapy are considered to have DTP irrespective of the cause of heart failure.

Comment 9: Dual antiplatelet therapy is class I recommended in ACS patients. Therefore, this should not be regarded as a DDI.

Response 9: Accommodated.

Comment 10: Expand Table 4 with patient characteristics, such as age, comorbidities, gender, renal function (eGFR), blood pressure and medication

Response 10: We have made a correction as per recommendation in Table 3. Except eGFR as renal function test is not routinely performed in the outpatient setting. 

Comment 11: How many patients have been hospitalized for HF in the past?

Response 11: Dear reviewer, I do have a data that showed only the number of hospitalized patients on the past one year. So I included the frequency (%) in Table 1. 

Comment 12: Please have the manuscript reviewed by a native English speaker.

Response12: Accommodated.

---

## [Decision Letter · Decision Letter 1]

4 Aug 2020

Evaluation of Drug Therapy Problems, Medication Adherence and Treatment Satisfaction among Heart Failure Patients on Follow-up at a Tertiary Care Hospital in Ethiopia

PONE-D-19-34755R1

Dear Dr. Berha,

We’re pleased to inform you that your manuscript has been judged scientifically suitable for publication and will be formally accepted for publication once it meets all outstanding technical requirements.

Kind regards,

Hans-Peter Brunner-La Rocca, M.D.

Academic Editor

PLOS ONE

Additional Editor Comments (optional):

Reviewers' comments:

Reviewer's Responses to Questions

**Comments to the Author**

1. If the authors have adequately addressed your comments raised in a previous round of review and you feel that this manuscript is now acceptable for publication, you may indicate that here to bypass the “Comments to the Author” section, enter your conflict of interest statement in the “Confidential to Editor” section, and submit your "Accept" recommendation.

Reviewer #1: All comments have been addressed

2. Is the manuscript technically sound, and do the data support the conclusions?

Reviewer #1: Yes

3. Has the statistical analysis been performed appropriately and rigorously? 

Reviewer #1: I Don't Know

4. Have the authors made all data underlying the findings in their manuscript fully available?

Reviewer #1: Yes

5. Is the manuscript presented in an intelligible fashion and written in standard English?

Reviewer #1: Yes

6. Review Comments to the Author

Reviewer #1: (No Response)

7. PLOS authors have the option to publish the peer review history of their article (what does this mean?). If published, this will include your full peer review and any attached files.

Reviewer #1: No

---

## [Editor Report · Acceptance letter]

17 Aug 2020

PONE-D-19-34755R1 

Evaluation of Drug Therapy Problems, Medication Adherence and Treatment Satisfaction among Heart Failure Patients on Follow-up at a Tertiary Care Hospital in Ethiopia 

Dear Dr. Berha:

I'm pleased to inform you that your manuscript has been deemed suitable for publication in PLOS ONE. Congratulations! Your manuscript is now with our production department. 

Kind regards, 

on behalf of

Dr. Hans-Peter Brunner-La Rocca 

Academic Editor

PLOS ONE